# Brain Response to Non-Painful Mechanical Stimulus to Lumbar Spine

**DOI:** 10.3390/brainsci8030041

**Published:** 2018-03-01

**Authors:** Zaid M. Mansour, Laura E. Martin, Rebecca J. Lepping, Saddam F. Kanaan, William M. Brooks, Hung-Wen Yeh, Neena K. Sharma

**Affiliations:** 1Department of Physical and Occupational Therapy, Hashemite University, Zarqa 13115, Jordan; zaid.modhi@hu.edu.jo; 2Hoglund Brain Imaging Center, University of Kansas Medical Center, Kansas City, KS 66160, USA; lmartin2@kumc.edu (L.E.M.); rlepping@kumc.edu (R.J.L.); wbrooks@kumc.edu (W.M.B.); 3Department of Preventive Medicine and Public Health, University of Kansas Medical Center, Kansas City, KS 66160, USA; 4Department of Rehabilitation Sciences, Jordan University of Science and Technology, Irbid 22110, Jordan; sfkanaan@just.edu.jo; 5Laureate Institute for Brain Research, 6655 South Yale Ave, Tulsa, OK 74136, USA; hyeh@laureateinstitute.org; 6Department of Physical Therapy and Rehabilitation Science, University of Kansas Medical Center, Kansas City, KS 66160, USA

**Keywords:** fMRI, mechanical pressure, lower back, brain activation pattern, pressure device

## Abstract

Pressure application to the lumbar spine is an important assessment and treatment method of low back pain. However, few studies have characterized brain activation patterns in response to mechanical pressure. The objective of this study was to map brain activation associated with various levels of mechanical pressure to the lumbar spine in healthy subjects. Fifteen healthy subjects underwent functional magnetic resonance imaging (fMRI) scanning while mechanical pressure was applied to their lumbar spine with a custom-made magnetic resonance imaging (MRI)-compatible pressure device. Each subject received three levels of pressure (low/medium/high) based on subjective ratings determined prior to the scan using a block design (pressure/rest). Pressure rating was assessed with an 11-point scale (0 = no touch; 10 = max pain-free pressure). Brain activation differences between pressure levels and rest were analyzed. Subjective pressure ratings were significantly different across pressure levels (*p* < 0.05). The overall brain activation pattern was not different across pressure levels (all *p* > 0.05). However, the overall effect of pressure versus rest showed significant decreases in brain activation in response to the mechanical stimulus in regions associated with somatosensory processing including the precentral gyri, left hippocampus, left precuneus, left medial frontal gyrus, and left posterior cingulate. There was increase in brain activation in the right inferior parietal lobule and left cerebellum. This study offers insight into the neural mechanisms that may relate to manual mobilization intervention used for managing low back pain.

## 1. Introduction

The brain’s role in the perception of mechanical pressure to the lumbar spine is not fully understood. Advances in neuroimaging techniques have expanded our knowledge of brain activation patterns in response to various sensory stimuli. Sensory stimuli such as, pressure, pain, touch, proprioception, and heat are associated with different sensory receptors throughout the body [1]. These different stimuli activate different regions of the brain including primary and secondary somatosensory cortices, cingulate, and insular cortex [2]. Mechanical pressure is a type of sensory stimulus that is applied to the lower back daily during activities such as sitting, walking, and lifting. Additionally, mechanical pressure is used clinically as a method of assessing and treating low back pain. However, few studies have examined the cortical representation of the lower back (lumbar spine) specifically with regards to mechanical stimuli.

Anatomical studies in nonhuman primates [3,4,5] and in other animals [6] have identified the medial aspect of the primary somatosensory cortex (S1; Brodmann areas 3,1,2) as representing the trunk region. Functional neuroimaging findings have reported activation in the primary somatosensory cortex in humans in response to electrical [7] and tactile stimulation (i.e., light touch) [8] delivered at anterior or posterior aspects of the trunk. The mechanosensory afferents from lumbar facet joints, ligaments, and discs respond mainly to mechanical stresses and are important for mobility, stability, and postural control [4,9]. Mechanical stresses are imparted on the lumbar spine during activities of daily living. They are also used during clinical physical examination to elicit patients’ response, since pressure on the spine can re-elicit painful symptoms. To date, the cortical response to mechanical stress has not been fully studied. To our knowledge, only two studies have examined brain activation responses to mechanical stimuli on the lower back in healthy subjects. The first study reported functional magnetic resonance imaging (fMRI) brain activation in primary and secondary somatosensory cortices (S1, S2), the inferior parietal, temporal, and insular cortices of the right hemisphere in response to an intense—yet non-painful—vibration stimulus to bilateral lower back in healthy subjects [10]. In the second study, light manual pressure applied to lumbar segments 1, 3, and 5 (L1, L3, and L5) in healthy subjects elicited activation in bilateral S1 and S2 and anterior cerebellum [11]. However, these studies are targeted to assess cortical response to vibration and light pressure only. The objective of the current study was to determine brain activation patterns in response to different levels of mechanical pressure on the lumbar spine in healthy subjects, since greater than minimum amount of pressure is placed on the spine during daily activities. Moreover, different levels of pressure are used for different purposes throughout clinical assessment and treatment [12]. We hypothesized that different pressure levels would lead to different brain activation patterns. Mapping out brain activation patterns to multiple pressure levels could improve clinical testing procedure and treatment based on pressure delivered to the lumbar spine, as this is a commonly applied manual technique for assessment and treatment of low back pain.

## 2. Materials and Methods

### 2.1. Participants

Fifteen right-handed healthy subjects (mean age, 33.46 ± 8.5 years; 10 males) were recruited from the University of Kansas Medical Center through mass e-mails and word of mouth. Exclusion criteria were (1) history of low back or lower limb pain within the past 6 months, (2) history of spinal surgery, (3) neurological, neuromuscular, or psychiatric disorders, (4) history of spinal cord compression, tumor, or infection, and (5) magnetic resonance imaging (MRI) exclusions: ferromagnetic metallic objects in the body, cardiac pacemaker, pregnancy, etc. The study was approved by the Institutional Review Board. All subjects gave written informed consent prior to their participation in the study and were compensated for their time and travel expenses.

### 2.2. Experimental Procedure

Each subject’s lumbar segment 3 (L3) was identified using the iliac crest as an anatomical landmark. The lumbar segment at the level of the iliac crest was considered L4 vertebrae (or interspace between L4 and L5). The skin was marked (with ink) and a belt with a cutout corresponding to the pressure device was placed around the subject’s waist to ensure correct stimulation of L3 in supine position. In the magnet, subjects laid supine over a foam board that had a cutout for delivery of the mechanical pressure. Mechanical pressure was applied to the L3 segment level using a custom-made pneumatic device constructed in our laboratory. Compressed air was delivered to an inflatable air bladder at varying pressures and frequencies timed by a computer software (E-Prime, Psychology Software Tools, Inc., Sharpsburg, PA, USA). We fixed two small plastic balls to the bladder to provide bilateral contact points approximately 2 cm lateral to midline. The height of the balls was adjusted to almost contact the skin surface to accommodate individual subjects’ lumbar curves (Figure 1). The entire unit was secured on a wooden stick that could be adjusted caudally and cephalically on the MRI table for accurate anatomic alignment. The device was positioned so that the balls were not in contact with the subject’s back when the air bladder was deflated. The belt was loosened after confirming accurate placement of the pressure device on the subject’s back. The subject’s head was secured with foam pads to minimize head movement. Both legs were supported in a partially flexed knee and hip position with a foam wedge and pillows under the knees to obtain a flat back posture.

Once in position (on the scanner table but not inside the magnet), the variable pressure conditions were optimized for each subject based on their subjective ratings. Three increasing, but non-painful, pressures were determined for each subject prior to the scan, starting at 5 pounds per square inch “psi” to the maximum pressure range of 30 psi. The device was calibrated by varying the pressure to obtain three consistent ratings per subject: low, medium, and high, using a 0–10 pressure rating scale where 0 = no pressure (no touch) and 10 = maximum pain-free pressure. Therefore, the pressure levels per subject were based on subjective ratings rather than pressure in pounds per square inch (psi) units. Across subjects, mean ratings (±SD) for low, medium, and high were 2.13 ± 0.91, 3.80 ± 0.86, and 5.40 ± 1.45, on the 0–10 pressure scale, respectively. Critical for our study, none of the pressure levels caused pain. After pressure device calibration and the subject’s experience of pressure delivery, subjects were pushed inside the magnet.

### 2.3. Block Design

Before the scan started, a test trial of the fMRI task was administered to familiarize subjects with the task. The examiner applied pressure during the scan using a block design (stimulus, rest). During the stimulus condition of the fMRI task, the air bladder was inflated causing the plastic balls to rise up against the subject’s lumbar spine, providing pressure to the bilateral aspect of spinous processes at L3. During the rest condition, the bladder was deflated to 0 psi and the balls were no longer in contact with the subject’s back. Six functional scans were obtained—two with low, two with medium, and two with high pressure—with pairs of selected consecutive runs utilizing the same pressure. The order of pressure level presentation was randomized across subjects. Each scan included an initial 28 s rest, and then 9 blocks (4 rest and 5 stimulus conditions), with each block consisting of either a 23.3 s rest period (no pressure) or a 17.6 s stimulus period (pressure) (Figure 2) making each scan 208 s long. The block design was presented to the examiner—but not to the subject—on a computer screen in the console room using E-Prime software. These visual cues allowed the examiner to identify rest and stimulus periods in order to manually control the pressure stimulus.

### 2.4. Data Acquisition and Analysis

All magnetic resonance (MR) images were acquired at 3T MR (Siemens Medical Solutions, Erlangen, Germany) at the Hoglund Brain Imaging Center, University of Kansas Medical Center. T1-weighted sagittal localizing series were acquired. High-resolution anatomic images were acquired using a T1-weighted MPRAGE sequence: TR = 2300 ms, TE = 3.05 ms, flip angle = 8°, matrix = 256 × 256 mm, and voxel size = 0.94 × 0.94 × 1.00 mm. After acquiring the structural sequence, six blood oxygen level-dependent (BOLD) scans were acquired for the functional block design task using the following parameters: TR = 2000 ms, TE = 50 ms, flip angle = 90°, and voxel size = 3.75 × 3.75 × 5.00 mm.

BOLD sequences were analyzed using AFNI software (Analysis of Functional NeuroImaging, Milwaukee, WI, USA). Preprocessing steps included time shifting, motion correction, alignment of the structural images to the functional images, warping to standard Talairach space, reslicing of the functional data to 3.50 × 3.50 × 3.50 mm, and spatial smoothing (using 6 mm full-width at half-maximum Gaussian). Individual time points were assessed for outliers. Time series were censored from the analysis if >50% of the voxels in the volume were considered outliers. In addition, time points plus one TR before and two TRs after were censored if motion was more than 0.30 mm between TRs.

The neural response to each of the pressure levels was estimated at the individual subject level using the general linear model. Regressors representing the three pressure levels, as well as the six motion-parameter estimates, were entered into the model. We used the restricted maximum likelihood (REML) approach implemented by AFNI’s 3dREMLfit to estimate the regressors of interest.

### 2.5. Whole-Brain Statistical Analysis

Mixed-model analysis of variance (ANOVA) was used for group level analysis. The independent variable was “pressure level” (low, medium, and high). Voxel intensity values were considered significant if the activation signal passed a false discovery rate (FDR) threshold (*p* < 0.05) in order to correct for multiple comparisons.

## 3. Results

### 3.1. Pressure Rating

Repeated measure ANOVA with Bonferroni correction for GroupWise comparison was conducted to determine differences in pressure amount and subjective rating. The *F*-test of absolute pressure, in psi, for each level was significantly different across all subjects (*F*_2,28_ = 398.22, *p <* 0.001). Low pressure mean was 13.33 ± 5.23, medium was 18.67 ± 5.16, and high was 24.00 ± 5.41 psi. Low pressure was significantly different than medium and high (*p* < 0.001); medium was significantly different than high (*p* < 0.001). Additionally, the subjective rating (0–10) for pressure levels was also significantly different (*F*_2,28_ = 88.78, *p <* 0.001). Low pressure rating mean was 2.13 ± 0.91; medium rating was 3.80 ± 0.86; and high rating was 5.40 ± 1.45. Low rating was significantly different than medium and high (*p* < 0.001); medium rating was different than high (*p* < 0.001).

### 3.2. Brain Activation

The mixed-model ANOVA revealed no significant main effect of pressure (no voxels survived FDR *p* < 0.05) indicating that individual pressure levels (low, medium, and high) did not result in significantly different brain activation patterns; therefore, all further analyses were collapsed across the three pressure levels. In comparison to rest (no pressure), decreased brain activation in response to mechanical pressure was observed in the bilateral precentral gyrus, bilateral postcentral gyri, left hippocampus, left precuneus, left medial frontal gyrus, and left posterior cingulate gyrus (Figure 3). The right inferior parietal lobule and the left cerebellum (cerebellar tonsil) showed increased activation during pressure compared to rest (Figure 4). Peak Talairach coordinates [13] in x, y, and z planes, cluster size, and the *t*-statistic of the peak voxel are presented in Table 1.

## 4. Discussion

We investigated brain activation patterns from a non-painful mechanical stimulus delivered to the lower lumbar region of healthy subjects in the supine position. We found that non-painful pressure applied to the lumbar spine resulted in decreases in brain activation in somatosensory regions and prefrontal regions and increases in brain activation in parietal and cerebellar regions. However, brain activation did not differ with varying pressure levels applied to the spinous process of L3 of healthy subjects, even though subjects rated these pressure levels differently. These results suggest that although healthy subjects were able to subjectively discriminate between different pressure levels the brain activation remained the same.

The regions that responded to the pressure in our study have been shown in several previous studies. The precentral gyrus is mainly involved in voluntary movements and the integration of sensorimotor information [14]. Nonetheless, several human and animal studies have shown that precentral gyrus is involved in pain modulation [15,16,17]. Kobayashi et al. reported increased activation in right premotor cortex in response to painful mechanical pressure to left lower back area in healthy subjects [18]. The posterior cingulate cortex is activated during cognitive function and emotional stimuli and regulates focus of attention [19]. The posterior cingulate also serves as a region in the default mode network [20]. Both the precentral gyrus and posterior cingulate have been reported to be activated in pain caused by heat and mechanical stimulus [10,18] but the current findings suggest they also respond to non-painful mechanical pressure. The hippocampus, an important limbic area involved in learning and memory, plays an essential role in pain regulation and modulation of stress [21,22]. Its involvement in non-painful mechanical pressure modulation has not been investigated previously. However, one study showed pressure-dependent increased hippocampal blood flow following noxious mechanical stimulation of various cutaneous areas in anesthetized rats [23]. Decreased brain activation in those regions as a result to mechanical pressure may indicate a therapeutic effect or “distraction” caused by the application of non-painful mechanical pressure, as it is often used in treatment of low back pain [24]. More research is needed to further examine this assumption.

Increased activation during pressure was observed in two regions, right inferior parietal lobule and left cerebellum. The inferior parietal lobule is involved in sensorimotor integration, including reaching, eye movements, hand and lower body movements, and coordinating multiple body part movements [25], and is also a part of the default mode network [26]. Its role in pressure stimulation is not clear, however increased activation within the inferior parietal lobule in response to mechanical pressure applied to the thumb of healthy participants and patients with fibromyalgia has been reported [27].

Finally, cerebellar activation was also noted in our findings, which is commonly reported in studies examining pain and pressure modulation [28,29,30]. Nonetheless, the role of the cerebellum in pain and pressure modulation and processing is still under investigation [31]. Some evidence suggests that the cerebellum contributes to pain and pressure modulation and the cognitive experience distinguishing between painful and non-painful stimuli [32]. In our study, the stimuli were designed to be non-painful. Similar findings were reported by Boendermaker et al. following light manual pressure to the lower back [11]. Activation of the cerebellum following mechanical pressure to the lumbar spine adds to the growing literature of the cerebellum’s role in response to pressure [11,27].

Brain regions showing activation (increased and decreased activation) in our study are consistent with regions reported by Boendermaker et al. Those regions included bilateral somatosensory cortices, cerebellum, and various subcortical regions of the sensorimotor network in response to a low manual pressure applied to the lower back while subjects laid prone in the scanner [11]. However, the activation patterns during the stimulus in our study resulted in both decreased and increased activation compared to rest. Decreased brain activation following mechanical pressure is contrary to Boendermaker’s findings, however our study design differs from theirs in several important ways, including different pressure application techniques (manual versus mechanical device), subjective pressure levels, no application of control stimulus, and the position of the subjects (prone vs. supine lying). In our study, subjects were lying supine on the scanner table. This provided significant sensory and tactile input ascending from the subject’s back area that was in contact with the table throughout the scanning session. Furthermore, none of our subjects reported pain while in the scanner or during pressure application. In addition, the pressure that was used in the Boendermaker’s study caused “spinal movement at the beginning of the range, free of resistance and preventing adverse effects” [11] as described in their paper, compared to higher levels of pressure in our study. Again, this might have contributed to the differences in results.

The cortical representation of the spine (specifically localized area of one segment, L3) in the sensory homunculus covers a small area of cortex compared to the representations for other body parts, such as the hand or face [33]. As subjects were able to differentiate between the levels of pressure, we expected to detect differences in brain activation. Nonetheless, in the current study we were unable to detect such differences in brain activation possibly due to the small size of the region. Moreover, in our design we based the three levels of pressure (low, medium, and high) on subjective ratings from each individual subject prior to starting the scan using a 0–10 pressure scale. This classification may have minimized the effect of “pressure” given that there was some overlap in pressure levels between subjects, and we acknowledge this limitation to our design. Some subjects rated higher psi levels (20–25) as “low” pressure (perceived) while others rated the same amounts of pressure as “high”. Another limitation is related to the pressure device. We were limited to 30 psi because that was the maximum capacity of the pneumatic pump. Higher levels may have been useful for subjects who rated higher levels of pressure as “low” or “moderate”. Given our limitations, our design has minimized most of the other limitations form previous studies.

To minimize limitations of delivering pressure to the low back during an MRI scanning session, we developed a custom-made pressure device that delivers posterior-to-anterior directed variable strength mechanical pressure to specific segments of the spine in supine position during fMRI acquisition. More significantly, it caused minimal to no head motion during the scan. The average maximum displacement for all subjects was 2.12 mm. As mentioned earlier, several studies have examined pressure on the spine but not without limitations. The supine position of subjects, the subjective pressure levels, and controlled frequency of pressure delivery allowed us to overcome many of the limitations.

Posterior-to-anterior pressure of various amounts is delivered during spinal mobilization, which is commonly used for assessment and treatment for patients with low back pain [34]. Different mechanical devices have been employed during MRI to elicit brain activity [18,35,36]. Kobayashi et al. utilized a syringe to apply air pressure to subjects while they were in prone position [18]. Boendermaker et al. applied manual pressure on subjects’ back, also while subjects were in prone position [11]. These studies in many ways mirror clinical application of mechanical pressure to the back and delivered mechanical pressure while the subjects were placed in prone position. However, prone lying during MRI acquisition may not be feasible for many reasons. Many subjects are not able to lie prone as it causes discomfort and potentially additional stress that may restrict subject recruitment and influence brain activation. In previous studies all subjects were given the same amount of pressure, although our data corroborate previous findings that people have different subjective thresholds for different pressure levels [37]. Our design minimized these limitations. Finally, it should be noted that previous studies [10,11,18] focused only on increased activation patterns and excluded analyses of deactivation from rest. Decreased activations may therefore have been present in previous studies but were not reported. Here we examined both increased and decreased activation in response to pressure.

Future studies can use this pressure device in patients with low back pain, or other pain-related conditions where mechanical pressure is desirable to understand the cortical response to mechanical pressure in people with pain. Future studies may also compare brain activation patterns between the application of mechanical pressure to the back region and peripheral tissues to gain a better understanding of central nervous system sensitization in various pain conditions. Finally, the device can be modified to determine the central nervous system effects of manual therapy treatment to the lower back, such as spinal mobilization where various amounts of pressure are applied to the spine to assess and treat the back [12].

## 5. Conclusions

Our study shows that non-painful mechanical pressure applied to the lumbar spine of healthy subjects can lead to decreased brain activation in several somatosensory regions, which might mimic the relieving effect caused by manual mobilization to the lumbar spine in lower back pain patients.

## Figures and Tables

**Figure 1 brainsci-08-00041-f001:**
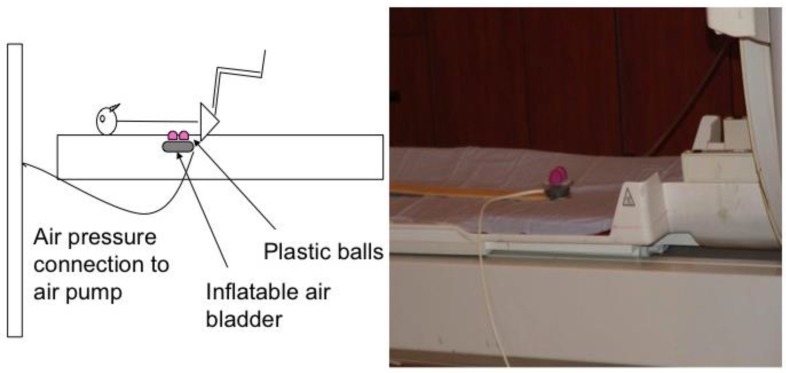
The pressure device.

**Figure 2 brainsci-08-00041-f002:**

The block design of the study. Red: pressure on 3.33 s; Blue: pressure off 3.33 s.

**Figure 3 brainsci-08-00041-f003:**
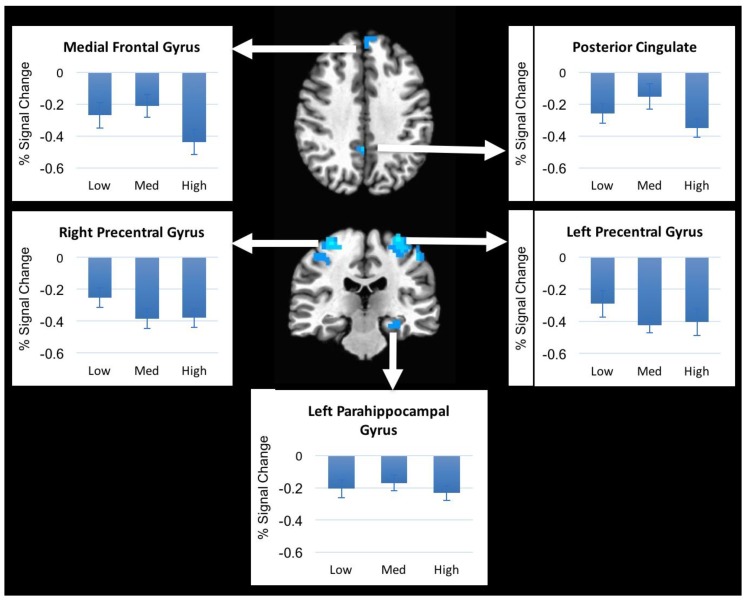
Decreased brain activation in response to mechanical pressure when compared to rest.

**Figure 4 brainsci-08-00041-f004:**
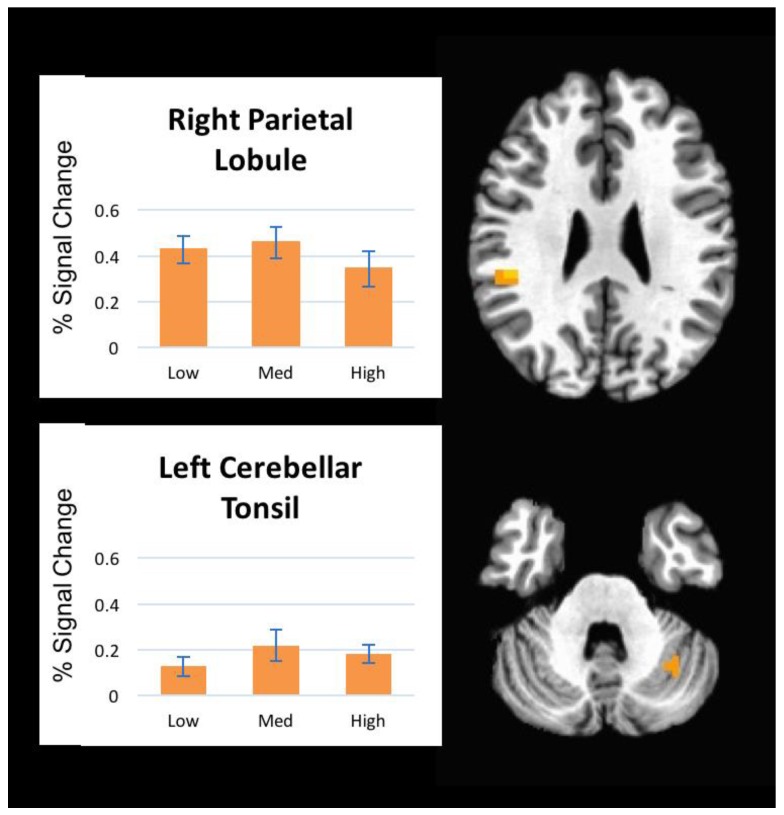
Increased brain activation in response to mechanical pressure when compared to rest.

**Table 1 brainsci-08-00041-t001:** Brain activation in response to mechanical pressure on the lumbar spine, FDR *p* < 0.05.

Name of Region	x Peak Coordinate	y Peak Coordinate	z Peak Coordinate	Cluster Size (mm^3^)	*t-*Stat (Max)
Lt Precentral Gyrus	−37	−15	59	2615	−8.91
Rt Precentral Gyrus	30	−22	59	1029	−8.65
Lt Posterior Cingulate	−9	−53	13	2229	−8.91
Lt Medial Frontal Gyrus	−5	52	34	686	−5.76
Lt Parahippocampal Gyrus	−26	−22	−11	600	−5.88
Rt Postcentral Gyrus	37	−25	48	600	−5.86
	33	−39	55	343	−6.32
Lt Postcentral Gyrus	−47	−22	45	343	−5.98
Rt Cingulate Gyrus	2	−46	38	514	−6.91
Lt Precuneus	−16	−64	48	428	−6.26
Rt Inferior Parietal Lobule	47	−32	27	557	7.30
Lt Cerebellar Tonsil	−33	−53	−32	214	5.68

x, y, and z coordinates are in the Talairach space, cluster size is in mm^3^.

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
