# Peer review of "Brain Response to Non-Painful Mechanical Stimulus to Lumbar Spine"

_brainsci, 2018, doi:10.3390/brainsci8030041_

Round 1

Reviewer 1 Report

The manuscript by Mansour and colleagues reports on brain activations (and de-activations) evoked by application of innocuous pressure to the lumbar spine in healthy subjects. The apparatus used to measure brain responses to low back pressure in the MRI scanner environment, while the subject is in a supine position, is an improvement over what has been tested previously. The application of pressure to this region is clinically relevant for the treatment of low back pain, so understanding how the brain responds to this type of stimulation in healthy individuals is important. Overall, the results are not as impactful as originally intended, since there were no significant differences between the three levels of pressure employed. The results of the pressure vs. off are somewhat difficult to interpret since they are mainly deactivations and they do not replicate previous findings in the literature that are fairly well accepted (e.g. increased activity in S1 during somatosensory stimulation). I am concerned that some aspects of the results are specific to certain features of the study design and not necessarily related to the perception of innocuous pressure on the lower back more generally. More specific comments on the manuscript are as follows:

1) If, as is depicted in Fig 2, each ON block began with 3.3 sec of the pressure not being applied, doesn’t this mean that the OFF blocks were technically 23.3 sec duration rather than 20? If the ON blocks are being modeled as starting 3.3 sec before the application of the first pressure pulse, this seems like a problem. I wonder if some significant positive activations would be uncovered if the ON blocks were modeled as beginning closer in time with the onset of the first pressure pulse and the offset of the 3rd (e.g. changing the first TR slices of each ON to OFF).

2) Why did the authors choose to do this alternating on-and-off pressure (3.3 sec duration) during the ON blocks instead of just applying a constant pressure? Since this stimulus is innocuous and is activating slowly adapting mechanoreceptors, I wouldn’t think that the perceived pressure would change much on the short term with a 20-sec stimulation. Having relatively long periods without pressure in the ON blocks is probably adding noise to the signal because there is no stimulus for half the time (you are mostly recording brain response to OFF during this time), and because the brain likely responds in some specific way to the removal of a pressure stimulus. See, for example, the literature on offset analgesia for thermal pain – the brain has ways of increasing the salience of changes in stimuli as they increase AND as they decrease.

Also, just from a practical standpoint, it seems difficult enough for the experimenter to precisely, manually turn the machine on and off for each 20-sec block, and even more difficult to time 4 sub-second sub-periods within the ON blocks. How confident are you in the accuracy and precision of the timing of this stimulus manipulation? Was there any readout about what the stimulator was doing during the scan that could be used to discard bad data?

3) Were any ratings of the stimuli obtained during or after the functional scans? It would be useful to know if people habituated to the pressure stimulus over the course of the six runs; if so, the evoked signals might have generally been stronger at the beginning of the session compared to the later runs.

4) There needs to be a more detailed description of the protocol for tailoring pressure levels to individual sensitivities. Were the authors trying to find pressures to elicit specific ratings on the 0-10 scale? What were they? Was the method of limits used or something else? How many stimuli were applied? What was the stimulus duration? Was the subject positioned in the scanner for this procedure? Were subjects trained at all prior to entering the scanner environment?

5) Section 2.4 seems out of place. Is there a reason this is not in the results section? Also, the authors should specifically state the statistical tests being used (e.g. repeated measures ANOVA or whatever post-hoc tests are reported). Maybe change section 2.4 to a short data analysis section instead.

6) Have the authors tried to analyze the data by modeling the high pressure ON blocks vs. the low pressure ON blocks? This could help get rid of some idiosyncrasies in the signals due to the specifics of the design (if above comments 1 and 2 are correct). Also, this would be a condition that would resemble some of the designs used previously in pain and touch imaging research, where there is a control stimulus for comparison of BOLD rather than no stimulus; the authors briefly mention this issue in the discussion. Order of runs should not be a confound here since it was randomized.

An alternative would be to model things similar to how the authors have now but to discard the medium runs from the analysis. The rationale would be that after a lack of significant findings in comparison of the 3 pressures, the authors then narrowed in on an analysis of the two pressures that were the most different from one another as a way of increasing signal differences between conditions. The medium pressure might not have been that perceptually different from either of the other two during the runs; the ratings, while significantly different at the beginning tailoring procedure, might not have been too different as people experienced the slight variations in pressure across the different functional scans.

7) Have the authors tried a more targeted approach (vs. the whole brain search) for studying the brain changes due to different pressure levels, like placing ROIs in S1 and other areas found to be activated previously?

Author Response

Comments and Suggestions for Authors

The manuscript by Mansour and colleagues reports on brain activations (and de-activations) evoked by application of innocuous pressure to the lumbar spine in healthy subjects. The apparatus used to measure brain responses to low back pressure in the MRI scanner environment, while the subject is in a supine position, is an improvement over what has been tested previously. The application of pressure to this region is clinically relevant for the treatment of low back pain, so understanding how the brain responds to this type of stimulation in healthy individuals is important. Overall, the results are not as impactful as originally intended, since there were no significant differences between the three levels of pressure employed. The results of the pressure vs. off are somewhat difficult to interpret since they are mainly deactivations and they do not replicate previous findings in the literature that are fairly well accepted (e.g. increased activity in S1 during somatosensory stimulation). I am concerned that some aspects of the results are specific to certain features of the study design and not necessarily related to the perception of innocuous pressure on the lower back more generally. More specific comments on the manuscript are as follows:

1) If, as is depicted in Fig 2, each ON block began with 3.3 sec of the pressure not being applied, doesn’t this mean that the OFF blocks were technically 23.3 sec duration rather than 20? If the ON blocks are being modeled as starting 3.3 sec before the application of the first pressure pulse, this seems like a problem. I wonder if some significant positive activations would be uncovered if the ON blocks were modeled as beginning closer in time with the onset of the first pressure pulse and the offset of the 3rd (e.g. changing the first TR slices of each ON to OFF).

Answer: In response to addressing this question, we reviewed our E-prime stimulus design and acknowledge that Fig 2 is incorrect.  We started each block with “on” followed by alternating on-off.  However, we have re-run the analysis to address the reviewers comment and moved the 3.3 sec at the end of the ON block where no stimulus was being applied to the OFF condition.  As now described in the paper the ON Blocks last 17.6 sec and the OFF Blocks last 23.3 sec. This change did not change the results.  A new figure (Fig 2) has been submitted displaying correct block design.

2) Why did the authors choose to do this alternating on-and-off pressure (3.3 sec duration) during the ON blocks instead of just applying a constant pressure? Since this stimulus is innocuous and is activating slowly adapting mechanoreceptors, I wouldn’t think that the perceived pressure would change much on the short term with a 20-sec stimulation. Having relatively long periods without pressure in the ON blocks is probably adding noise to the signal because there is no stimulus for half the time (you are mostly recording brain response to OFF during this time), and because the brain likely responds in some specific way to the removal of a pressure stimulus. See, for example, the literature on offset analgesia for thermal pain – the brain has ways of increasing the salience of changes in stimuli as they increase AND as they decrease.

Also, just from a practical standpoint, it seems difficult enough for the experimenter to precisely, manually turn the machine on and off for each 20-sec block, and even more difficult to time 4 sub-second sub-periods within the ON blocks. How confident are you in the accuracy and precision of the timing of this stimulus manipulation? Was there any readout about what the stimulator was doing during the scan that could be used to discard bad data?

Answer: We conducted an on-off design to model clinical application of spinal mobilization.  Spinal mobilization is a standard assessment and treatment method used in rehabilitation settings by clinicians (i.e. physical therapists and chiropractors), in which a pressure is directly applied on the spinous process in an oscillatory fashion with back and forth movement.  The depth of the mobilization could be shallow or deep, depending on stage of tissue healing and irritability. Greater depth or sustained pressure, when applied directly on the spinous processes or adjacent to it - as performed in our study design – could become painful.  To avoid potential noxious sensation and to replicate clinical application, we chose to perform on-off stimulus.   

Secondly, we have applied similar experimental design in other studies also and feel confident that we were precise in the application of stimulus and on-off paradigm.  The application of on-off design was practiced by the experimenter prior to data collection, and the trained experimenter was able to perform this design. Nevertheless the experiment was only machine-guided and not machine-operated; thus this is a limitation in the delivery of stimulus.  Given the nature of the BOLD response, we do not anticipate that small millisecond discrepancies in turning on and off would have had a significant impact on the overall results.

3) Were any ratings of the stimuli obtained during or after the functional scans? It would be useful to know if people habituated to the pressure stimulus over the course of the six runs; if so, the evoked signals might have generally been stronger at the beginning of the session compared to the later runs.

Answer: The rating was obtained prior to the scan and we did not collect rating throughout the scan or after the scan. Thus we are unable to test whether subjects habituated in terms of a subjective rating of pressure.  Furthermore, pressure delivery was counterbalanced across participants so that some participants received high pressure first whereas others received low or medium pressure first. This was done to minimize potential order effects related to the pressure level.  One limitation of this approach is we cannot look at habituation effects by examining diminished brain activation over time.

4) There needs to be a more detailed description of the protocol for tailoring pressure levels to individual sensitivities. Were the authors trying to find pressures to elicit specific ratings on the 0-10 scale? Find specific rating only to classify them into low, med and high What were they? Was the method of limits used or something else? How many stimuli were applied? What was the stimulus duration? Was the subject positioned in the scanner for this procedure? Were subjects trained at all prior to entering the scanner environment?

Answer: The aim of the pressure rating was to determine individual participant’s perceived or subjective pressure ratings and tailor the low, medium and high objective pressures (PSI) applied during the scan to the participant.  To determine the subjective pressure levels we set the participants on the table, explained the study design, applied 5 psi and determined if they experienced any pressure to their back on a scale of 0-10. If they reported 0, we went up to 10 psi and asked them to rate their pressure, once they felt the stimulus device (e.g. experienced low pressure) we adjusted the pressure amount accordingly to find their range from low to medium to high pressure based on their subjective rating.

Each subject experienced the pressure delivery during the optimization process described above and PSI level was determined according to their rating prior to the start of the scan. They were not trained on the task or block design, but they gained experience with the pressure device so they would know what to expect during the scan.  We intentionally did not want to train subjects for the order or amount of pressure delivered in order to minimize participants’ anticipation across runs.  

More detailed description of the protocol has been added in the manuscript.  

5) Section 2.4 seems out of place. Is there a reason this is not in the results section? Also, the authors should specifically state the statistical tests being used (e.g. repeated measures ANOVA or whatever post-hoc tests are reported). Maybe change section 2.4 to a short data analysis section instead.

Answer: Section 2.4 has now been moved to the results; it has been shorten, and the statistical tests are described.

6) Have the authors tried to analyze the data by modeling the high pressure ON blocks vs. the low pressure ON blocks? This could help get rid of some idiosyncrasies in the signals due to the specifics of the design (if above comments 1 and 2 are correct). Also, this would be a condition that would resemble some of the designs used previously in pain and touch imaging research, where there is a control stimulus for comparison of BOLD rather than no stimulus; the authors briefly mention this issue in the discussion. Order of runs should not be a confound here since it was randomized.

An alternative would be to model things similar to how the authors have now but to discard the medium runs from the analysis. The rationale would be that after a lack of significant findings in comparison of the 3 pressures, the authors then narrowed in on an analysis of the two pressures that were the most different from one another as a way of increasing signal differences between conditions. The medium pressure might not have been that perceptually different from either of the other two during the runs; the ratings, while significantly different at the beginning tailoring procedure, might not have been too different as people experienced the slight variations in pressure across the different functional scans.

Answer: We appreciate this perspective and examined the High vs. Low pressure levels in the whole brain analysis but no regions showed an effect of pressure.  We have also added bar graphs to Figures 3 and 4 to illustrate the pattern of brain responses across pressure levels.

7) Have the authors tried a more targeted approach (vs. the whole brain search) for studying the brain changes due to different pressure levels, like placing ROIs in S1 and other areas found to be activated previously?

Answer: To address this suggestion, we completed a targeted ROI approach by defining spherical ROIs around the peak coordinates in the S1 and SMA regions identified in respond to mechanical pressure according to the methods described in the article by Boendermaker et al 2014. Mean percent signal change was extracted from these ROIs and paired t-tests comparing Low vs. High pressure as well as the full ANOVA comparing all pressure levels were run in SPSS.  No significant results were found (p > .1).

Submission Date

21 December 2017

Date of this review

02 Jan 2018 18:08:39

Reviewer 2 Report

Summary: The authors presented a well-written manuscript assessing brain activation patterns using fMRI when a non-painful mechanical pressures (low medium and high pressure) were applied to the third segment of the lumbar spine (L3). Pressure in this location is viewed as both an assessment and treatment for lower back pain. Brain activation patterns were also compared to periods in which pressure was not present.

Comments:

1.     When discussing the 15 participants in line 72, the mean age is reported. Is this just for the 10 males that were mentioned or for all subjects? If this is just for the men, what is the mean age for the 5 women? Please clarify.

2.     If the low pressure mean and low rating scores were less than medium and medium less than high, it is redundant to mention that the low scores were less than high. This can be removed.

3.     Please clarify which post-hoc test was used in the statistics, as some are more conservative than others.

4.     The design block is confusing. In the text it is mentioned that pressure is applied for 20 seconds with 20 seconds of rest after. However the image provided details that the 20-second of applied pressure block has times where pressure is not applied. Equaling 3.33 seconds on and 3.33 seconds off. Please specify this more clearly within the methods section. If the pressure device is being manually controlled, is 3.33 seconds long enough to get a brain response adequately? Many studies have a longer paradigm. It would be interesting to hear a discussion on this.

5.     Figure 3 shows areas that are not listed within the text (i.e the somatosensory cortex). Did the results show this? Please clarify.

6.     How were brain areas chosen for Table 1? There are many missing areas that were stated in the text. It may be better to limit the areas in the text that will be discussed later.

7.     In line 201 there is mention of the medial temporal lobe, however, there is no further discussion of this in the paragraph. This sentence should be moved or eliminated all together.

Minor comments:

1.     Although there is discussion on the limitation of the psi of the device, it would be interesting to know if those that rated 20-25 psi as a low pressure were able to experience a medium and high pressure as the device was limited to 30 psi.

2.     There are a few minor grammatical issues that may cause confusion to the reader (i.e. line 248, 249 and 287). 

Author Response

The University of Kansas Medical Center

School of Health Professions

Physical Therapy and Rehabilitation Sciences

February 15, 2018

To:       Rena Wu, PhD

            Assistant Editor

            Brain Sciences Journal

Reviewer 2

Summary: The authors presented a well-written manuscript assessing brain activation patterns using fMRI when a non-painful mechanical pressures (low medium and high pressure) were applied to the third segment of the lumbar spine (L3). Pressure in this location is viewed as both an assessment and treatment for lower back pain. Brain activation patterns were also compared to periods in which pressure was not present.

Comments:

1.     When discussing the 15 participants in line 72, the mean age is reported. Is this just for the 10 males that were mentioned or for all subjects? If this is just for the men, what is the mean age for the 5 women? Please clarify.

Answer: the mean represent the entire sample size.  We made a change in the manuscript and moved the mean value next to the subjects to make it more clear.    

2.     If the low pressure mean and low rating scores were less than medium and medium less than high, it is redundant to mention that the low scores were less than high. This can be removed.

Answer: We agree and have made this change in the manuscript.

3.     Please clarify which post-hoc test was used in the statistics, as some are more conservative than others.

Answer: We used repeated measures ANOVA with Bonferroni correction for GroupWise / pairwise comparison.  We have included this information in the manuscript.

4.     The design block is confusing. In the text it is mentioned that pressure is applied for 20 seconds with 20 seconds of rest after. However the image provided details that the 20-second of applied pressure block has times where pressure is not applied. Equaling 3.33 seconds on and 3.33 seconds off. Please specify this more clearly within the methods section. If the pressure device is being manually controlled, is 3.33 seconds long enough to get a brain response adequately? Many studies have a longer paradigm. It would be interesting to hear a discussion on this.

Answer: As described above, this 3 sec ON/OFF alteration during the Stimulus blocks was deliberate to mimic the clinical environment.  Due to the nature of the hemodynamic response which lasts approximately 12 sec the 3 sec alteration should result in a sustained increase in activation over the 17 sec Stimulus Block. This design has been used in previous visual and motor stimulus block designs.

5.     Figure 3 shows areas that are not listed within the text (i.e the somatosensory cortex). Did the results show this? Please clarify.

Answer: Regions that were not explicitly listed in the text but were present in Figure 3 is due to these regions being part of a larger cluster of activation that spread from parietal to precentral gyrus.  Specifically, the text and Table 1 focused on where the peak activation within a cluster was present.  In order to address this issue we now corrected for multiple comparisons using an false discovery rate (FDR) approach to identify voxels that pass q < .05.  This approach identifies highly significant voxels as oppose to clusters of voxels that show similar patterns.  By implementing this approach we have been able to identify more discrete brain responses that are highly significant and do not spread across multiple brain regions. As seen in new Figures 3 and 4 the regions identified in the whole-brain analysis are similar to those shown in the previous version of the analysis. New Table 1 presents the peak voxels in each region and now better describes the regions that responded to mechanical pressure.

6.     How were brain areas chosen for Table 1? There are many missing areas that were stated in the text. It may be better to limit the areas in the text that will be discussed later.

Answer: Brain areas in Table 1 were identified in the whole brain analysis (i.e. regions that passed statistical significance for Pressure ON vs. OFF). As described above we are now using an FDR approach to identify more discrete regions as oppose to large clusters of activation that spread across multiple regions.   

7.     In line 201 there is mention of the medial temporal lobe, however, there is no further discussion of this in the paragraph. This sentence should be moved or eliminated all together.

 Answer: This sentence has been removed.

1.     Although there is discussion on the limitation of the psi of the device, it would be interesting to know if those that rated 20-25 psi as a low pressure were able to experience a medium and high pressure as the device was limited to 30 psi.

Answer: The goal of the current paper was to understand brain responses to objectively defined pressure levels.  Therefore, the analysis focused on the PSI defined pressure levels and our behavioral data supported this approach showing a main effect of pressure based on objective ratings (i.e. PSI) and subjective ratings (i.e. participants’ self-reported pressure rating) obtained prior to the start of the scan. However, based on this comment we did go back and rerun the analysis based on subjective ratings and found no brain regions that showed a differential response to subjective pressure ratings.

2.     There are a few minor grammatical issues that may cause confusion to the reader (i.e. line 248, 249 and 287). 

Answer: We have made these corrections. 

Round 2

Reviewer 1 Report

The authors have adequately addressed my concerns.

Reviewer 2 Report

Thank you, all comments were addressed. Congratulations on your manuscript!